# Molecular Characterization of the MoxR AAA+ ATPase of *Synechococcus* sp. Strain NKBG15041c

**DOI:** 10.3390/ijms25189955

**Published:** 2024-09-15

**Authors:** Kota Mano, Kentaro Noi, Kumiko Oe, Takahiro Mochizuki, Ken Morishima, Rintaro Inoue, Masaaki Sugiyama, Keiichi Noguchi, Kyosuke Shinohara, Masafumi Yohda, Akiyo Yamada

**Affiliations:** 1Department of Biotechnology and Life Science, Tokyo University of Agriculture and Technology, Tokyo 184-8588, Japan; kota.mano@yohda.net (K.M.); k.noi@people.kobe-u.ac.jp (K.N.); kumiko.oe@yohda.net (K.O.); takahiro.mochizuki@yohda.net (T.M.); k_shino@cc.tuat.ac.jp (K.S.); 2Institute for Integrated Radiation and Nuclear Science, Kyoto University, Osaka 590-0494, Japan; morishima.ken.8e@kyoto-u.ac.jp (K.M.); inoue.rintaro.5w@kyoto-u.ac.jp (R.I.); sugiyama.masaaki.5n@kyoto-u.ac.jp (M.S.); 3Instrumentation Analysis Center, Tokyo University of Agriculture and Technology, Tokyo 184-8588, Japan; knoguchi@cc.tuat.ac.jp

**Keywords:** chaperone, cyanobacteria, stress, MoxR, analytical ultracentrifugation

## Abstract

We isolated a stress-tolerance-related gene from a genome library of *Synechococcus* sp. NKBG15041c. The expression of the gene in *E. coli* confers resistance against various stresses. The gene encodes a MoxR AAA+ ATPase, which was designated SyMRP since it belongs to the MRP subfamily. The recombinant SyMRP showed weak ATPase activity and protected citrate synthase from thermal aggregation. Interestingly, the chaperone activity of SyMRP is ATP-dependent. SyMRP exists as a stable hexamer, and ATP-dependent conformation changes were not detected via analytical ultracentrifugation (AUC) or small-angle X-ray scattering (SAXS). Although the hexameric structure predicted by AlphaFold 3 was the canonical flat-ring structure, the structures observed by atomic force microscopy (AFM) and transmission electron microscopy (TEM) were not the canonical ring structure. In addition, the experimental SAXS profiles did not show a peak that should exist in the symmetric-ring structure. Therefore, SyMRP seems to form a hexameric structure different from the canonical hexameric structure of AAA+ ATPase.

## 1. Introduction

All organisms on Earth, including microorganisms, sustain growth under various environmental stresses by expressing mechanisms of resistance to them. In nature, a diverse range of organisms have evolved resistance to environmental stresses such as high temperature, low or high pH, and high salt concentrations. Some organisms are capable of growing in harsh environments because they possess genes that confer resistance mechanisms against external environmental stresses. Consequently, efforts have been made to explore stress tolerance genes based on the genomic information of such organisms [1]. Stress often induces denaturation of proteins. Therefore, under stress conditions, the expression of a group of proteins called molecular chaperones is induced, which helps mitigate the effects of various stresses. Many molecular chaperones are heat shock proteins that are responsible for maintaining the proper folding of proteins inside cells [2,3]. Molecular chaperones not only assist in the folding of newly synthesized polypeptides but also perform various functions in intracellular protein quality control, such as refolding denatured proteins, preventing aggregation, facilitating transport, and degrading proteins. Molecular chaperones play essential roles in all processes, from protein synthesis to degradation.

*Synechococcus* sp. strain NKBG15041c was isolated as a fast-growing marine cyanobacterium from coastal seawater in Okinawa Prefecture in Japan [4]. This strain is more resistant to thermal and acid stress than other cyanobacteria, such as *Synechocystis* sp. PCC6803. *Synechococcus* sp. strain NKBG15041c could grow in medium at pH 4 for 4 days. In contrast, *Synechocystis* sp. PCC 6803 was unable to grow in acidic media. The draft genome sequence of *Synechococcus* sp. strain NKBG15041c has already been reported [5]. Previously, we established a functional screening method in which *E. coli* was used to obtain the genes responsible for stress resistance [6]. Using this method, we isolated a gene associated with stress resistance from the cDNA library of *Synechococcus* sp. strain NKBG15041c. A clone that presented increased acid tolerance contained a gene encoding the MoxR AAA+ ATPase.

MoxR AAA+ ATPases are divided into nine subfamilies (MoxR proper (MRP), TM0930, RavA, KCR_1472, GvpN, CbbQ/NorQ, YehL, PA2707, and CoxD) [7]. They are widespread among bacteria and archaea but have not yet been detected in eukaryotes. The cellular functions of MoxR AAA+ ATPases are not well characterized. Some MoxR AAA+ ATPases are proposed to have chaperone-like functions for the maturation of specific protein complexes or for the insertion of cofactors into proteins. CbbQ, a member of the MoxR family of ATPases, has been shown to activate ribulose-1,5-bisphosphate carboxylase/oxygenase (RubisCo) together with CbbO [8,9]. RavA, together with ViaA, is functionally associated with anaerobic respiration in *E. coli* through interactions with the fumarate reductase electron transport complex [10,11]. Each of these functions, in terms of structure, requires the formation of homo-oligomers, especially in a hexameric ring structure. In the Gram-negative bacterium *Francisella tularensis*, promoter region analysis suggested that the expression of MoxR family ATPases is likely regulated by the heat-shock-related transcriptional regulator σ32. It has also been reported that the deletion of the MoxR family ATPase led to a mutant bacterium with increased vulnerability to various stress conditions, including oxidative and pH stresses, suggesting that these genes play important roles in the stress tolerance mechanism of the organism. However, a detailed functional analysis has not been conducted [12]. In this study, we characterized the structure and function of the MoxR family of proteins, which confer stress resistance to *E. coli* and seem to be responsible for the high stress resistance of *Synechococcus* sp. strain NKBG15041c.

## 2. Results

### 2.1. Screening of a Gene Responsible for the Stress Tolerance of Synechococcus *sp.* Strain NKBG15041c

We constructed a genomic DNA library of *Synechococcus* sp. strain NKBG15041c via the pBluescript SK-vector. *E. coli* cells were transformed with the DNA library, and the transformants were selected via acid treatment. Clones were selected for primary screening by acid treatment in saline solution at pH 3.0 for 10 min with shaking. The secondary screening was performed using the same procedure applied to the clones obtained in the primary screening. Among the transformants, one clone presented significantly high salt resistance. The inserted gene was sequenced and found to encode a putative protein of 331 amino acids. This protein was identical to the methanol dehydrogenase regulatory protein MoxR-like ATPase of *Synechococcus* sp. NIES-970 except for the N-terminal extension. Since there is no putative ribosome binding site upstream of the second ATG, the protein should express with the N-terminal extension; it also shows significantly high homology with a MoxR AAA+ ATPase of Synechocystis sp. PCC 6803 (NCBI-ProteinID: BAA10517, CyanoBase: slr0835) (Figure 1a), except for the N-terminal extension. These proteins belong to the MRP subfamily of MoxR AAA+ ATPases. Therefore, we named the encoded protein SyMRP (Figure 1b).

The coding sequences for SyMRP and BAA10517 were amplified by PCR and then inserted into the pBluescript SK-vector. These constructs were designed to express the proteins under the lac promoter. Compared with *E. coli* transformed with an empty vector, *E. coli* expressing SyMRP presented significant acid, salt, and heat stress tolerance (Figure 2). Interestingly, *E. coli* expressing BAA10517 presented relatively weak tolerance of stress. We speculated that SyMRP functions as a molecular chaperone both in *Synechococcus* sp. NIES-970 and in *E. coli*. To confirm this, we examined the structure and function of SyMRP in vitro.

### 2.2. Expression and Functional Characterization of SyMRP

SyMRP with a His-tag at the C-terminus was expressed in *E. coli* and purified using affinity chromatography and ion exchange chromatography. As SyMRP is an AAA+ ATPase, we first assessed its ATPase activity. SyMRP exhibited ATPase activity, with its relative activity at 37 °C estimated to be approximately 10.9 nmol Pi/min/mg protein. To investigate the ATPase activity of SyMRP under stress conditions, we measured its activity at 42 °C, a temperature typically associated with the induction of heat stress in cyanobacteria [13,14]. The ATPase activity at 42 °C was approximately 136 nmol Pi/min/mg protein, which was 12.5 times greater than that at 37 °C.

We subsequently examined chaperone ability via a thermal aggregation assay using porcine citrate synthase (CS). CS was incubated at 43 °C with or without SyMRP, and the light scattering intensity at 500 nm was measured to monitor the thermal aggregation process (Figure 3). The addition of SyMRP had almost no effect on the thermal aggregation of CS. However, in the presence of ATP, SyMRP protected CS from thermal aggregation. The effect increased with increasing SyMRP. ATP itself is known to partly suppress the thermal aggregation of CS [15]. However, the effect was less pronounced than that in the presence of SyMRP. The addition of ADP had no effect. Therefore, we concluded that SyMRP is an ATP-dependent molecular chaperone.

### 2.3. Oligomeric Structure of SyMRP

The molecular architecture of SyMRP was examined by size exclusion chromatography (SEC) and analytical ultracentrifugation (AUC). In SEC, it appears as a single peak with an apparent molecular mass of 236 kDa (Appendix A). Considering the molecular mass of the SyMRP monomer (36 kDa), SyMRP forms hexamers similar to the other MoxR AAA+ ATPases. Furthermore, the molecular mass of the main component at *s*_20,w_ = 7.60 S in the AUC profile was calculated to be 221 kDa, corresponding to the hexamer. The sedimentation coefficient distribution revealed by the AUC was almost unchanged by the addition of ATP (Figure 4 and Table 1).

We then tried to capture the ATP-dependent conformational change in SyMRP with SAXS [16,17]. As shown in Figure 5, the SAXS profile for SyMRP exhibited an increase in the low *q* range (*q* ≤ 0.025 Å^−1^) due to large aggregates. We therefore focused on the higher *q* range (*q* > 0.025 Å^−1^), where the contribution of the large aggregates was relatively small. Unexpectedly, there was almost no change in the SAXS profile between samples with and without ATP (Figure 5). This finding indicates that ATP does not affect the conformation of the SyMRP hexamer. Therefore, its ATP-dependent chaperone activity should be due to local structural changes.

Figure 6 shows the model structure of SyMRP predicted by AlphaFold 3. The predicted hexameric structure is the canonical flat ring. The N-terminal extensions seem to be propellers connected to the ring. There is a large central pore in the hexameric ring. The hexameric structure is not affected by the presence of ATP or ADP with Mg ions.

We then observed the oligomeric structures of SyMRP via AFM and EM. In AFM, small and large oligomers were observed. The small ones seem to be dimers or trimers (Figure 7a). The large hexamers seem to be hexamers (Figure 7b). Unexpectedly, the hexameric structure observed by AFM was a complex of three dimers (Figure 7b). Since the small oligomer, probably the dimer, was also detected as a minor component by AUC measurement, these results suggested that a hexameric structure may be formed using a dimer as the minimum number of units. The TEM image seems to be a flat-ring structure. But its resolution was too poor for observing C6 symmetry (Figure 7b). Since the experimental SAXS profiles also did not show a peak at *q*~0.2 Å^−1^ (the solid line and arrow in Figure 5), which is predicted from the symmetric-ring structure proposed by AlphaFold 3, the hexamer is expected to have a less symmetrical or dynamically fluctuating conformation in solution.

## 3. Discussion

MoxR family ATPases often function in collaboration with other proteins. These proteins have a von Willebrand factor A (VWA) domain, which has been implicated in protein–protein interactions. In a recent study, proteins belonging to seven subfamilies, excluding KCR_1472 and the GvpN subfamily, were associated with proteins with VWA domains [7]. These proteins are often encoded in close genomic proximity to the MoxR family ATPase gene. VWA proteins belonging to the MxaL/MxaC, DP0636, and other MRP-related groups are thought to act in concert with the MRP subfamily [7,18].

Recently, the complex structure of NorQ/NorD chaperones from *Paracoccus denitrificans* was determined [19]. NorQ, a member of the MoxR class of AAA+ ATPases, and NorD, a protein containing a VWA domain, are essential for nonheme iron (FeB) cofactor insertion into cytochrome c-dependent nitric oxide reductase. NorQ forms a circular hexamer with a monomer of NorD binding both to the side and to the central pore of the NorQ ring. Guided by AlphaFold predictions, the authors assigned the density that “plugs” the NorQ ring pore to the VWA domain of NorD, with a protruding “finger” inserting through the pore, and suggested this binding mode to be general for MoxR/VWA couples. Similar to NorQ, there is a large central pore in the hexameric structure of SyMRP. Although several genes encoding VWA domain-containing proteins were found in the genome of *Synechococcus* sp. NKBG15041c, their genomic locations are far from those of SyMRP. Thus, identification of the functional partner protein is difficult.

SyMRP confers resistance to *E. coli* against various stresses. It also protects CS from thermal aggregation in vitro. We speculate that SyMRP functions as a general chaperone without collaboration with VWA domain-containing proteins. Interestingly, its chaperone function depends on ATP. Although we expected a large conformational change due to ATP binding, the AUC and SAXS analyses revealed almost no change with and without ATP. Curiously, the effects of a homolog, BAA10517, from *Synechocystis* sp. PCC6803 on CS aggregation were relatively moderate. The large difference between SyMRP and BAA10517 is found in the N- and C-terminal extensions. Thus, the N- or C-terminal extension might be related to its chaperone activity.

Wang et al. conducted a proteome analysis of the model cyanobacterium *Synechocystis* sp. PCC 6803 [20]. BAA10517 was found to localize to all fractions in the cell, plasma membrane, outer membrane, and cytoplasm. This result suggests that SyMRP functions as a chaperone that contributes to the stabilization of a wide range of proteins. Previous studies have shown that SyMRP is expressed more under stress conditions than under nonstress conditions, indicating the need for further proteome analysis under stress conditions in the future.

The hexameric structure modeled by AlphaFold 3 was a canonical hexameric flat-ring structure. Almost no structural change was predicted by the addition of ATP to the model. However, curiously, such a canonical hexameric ring structure was not observed by AFM or TEM. The SAXS data also did not support such a symmetric-ring structure. Since all experimental data support AlphaFold 3 prediction, we think that SyMRP does not take the symmetric-ring structure. Therefore, X-ray or CryoEM structures should be determined to reveal the molecular mechanism of SyMRP.

## 4. Materials and Methods

### 4.1. Strains and Media

*Synechococcus* sp. NKBG 15041c was grown in BG11 medium supplemented with 3% NaCl (*w*/*v*). *Synechocystis* sp. PCC 6803 was grown in BG11 medium.

### 4.2. Genomic DNA Library Construction

The genomic DNA of *Synechococcus* sp. NKBG 15041c and *Synechocystis* sp. PCC 6803 was extracted via a bead-beating method. Algal cells, harvested by centrifugation, were suspended in 200 μL of TE buffer, to which 200 μL of TE-saturated phenol and 0.3 g of glass beads were added. The suspension was then subjected to bead beating via a Multi Beads Shocker (MB601U(S), Yasui Kikai, Osaka, Japan). After centrifugation, the supernatant was recovered, and residual phenol was removed by mixing with an equal volume of chloroform. After RNase treatment, genomic DNA was obtained by ethanol precipitation.

The genome of *Synechococcus* sp. NKBG 15041c was partially digested by Sau3AI. The digested DNA fragments of 3–8 kDa were cloned and inserted the BamHI site of the pBluescript SK-vector. *E. coli* DH5α cells were subsequently transformed with the prepared DNA library.

### 4.3. Evaluation of Stress Tolerance in E. coli Expressing SyMRP and BAA10517

The SyMRP gene was amplified by PCR from the clone as a template using the primers 5′-TATGAGCTCGATCAAGATGGCGATCGC-3′ and 5′-TAGGTACCCTATTTTTCGGTCGTTGGTAAG-3′ and then inserted into the SacI/KpnI site of pBluescript SK- (pSK-SyMRP). The full-length BAA10517 gene was amplified via PCR from the genome of *Synechocystis* sp. PCC 6803 using the primers 5′-TATGGTACCATGCGAGAACAGATTTCTGT-3′ and 5′-TAGACTAGTTTAAATGGGTACGCTGTCCA-3′. The amplified DNA was inserted into the SacI/KpnI site of pBluescript SK- (pSK-BAA10517).

*E. coli* (DH5α) cells were transformed with pSK-SyMRP, pSK-BAA10517, or an empty pBluescript SK-vector as a control. The transformed cells were precultured in 2xYT supplemented with 50 μg/mL ampicillin liquid medium for approximately 12 h at 37 °C with shaking at 170 rpm. The bacterial suspension was diluted in the same liquid medium to achieve an OD_600_ of 1.0. A total of 1 mL of this bacterial suspension was added to 9 mL of 2xYT (50 μg/mL ampicillin) liquid medium to obtain an OD_600_ of 0.1 and then subjected to shaking cultivation under stress conditions of 1.0 M NaCl, pH 4.0, and 45 °C. Cultivation was performed in triplicate for each condition, and OD_600_ measurements were taken at regular intervals using a Ratio Beam Spectrophotometer U-5100 (Hitachi High-Tech, Tokyo, Japan). Additionally, after preculture, the bacterial suspension was diluted to OD600 values of 10^−1^, 10^−2^, 10^−3^, and 10^−4^ in 2YT liquid medium, and 10 μL of each dilution was spotted onto 2xYT agar plates containing 50 μg/mL ampicillin, as well as 2xYT agar plates supplemented with 0.8 M NaCl. Similarly, heat-treated (50 °C for 30 min) bacterial suspensions were spotted onto 2xYT agar plates and cultivated at 37 °C for approximately 15–20 h.

### 4.4. Expression and Purification of SyMRP

The SyMRP gene was amplified via PCR using the primers 5′-TATTCATATGGCGATCGCGCC-3′ and 5′-TATCCTCGAGTTTTCGGTCGTTG-3′ and then inserted into the NdeI/Xho site of the pET23b vector (pET-SyMBP). SyMRP with a His-tag at the C-terminus was expressed in *E. coli* BL21 (DE3) transformed with pET-SyMBP. SyMRP was purified by affinity chromatography using Ni Sepharose™ 6 Fast Flow (Cytiva); after dialysis, it was further purified by anion exchange chromatography (RESOURCE Q 6 mL (Cytiva, Marlborough, MA, USA)). The transformant was grown in BG11 medium supplemented with 3% NaCl (*w*/*v*). *Synechocystis* sp. PCC 6803 was grown in BG11 medium.

### 4.5. ATPase Activity

SyMRP was dissolved in 600 μL of TKM buffer (50 mM Tris-HCl, pH 7.5, 100 mM KCl, and 25 mM MgCl_2_) to a final concentration of 0.2 μM. To remove bound ATP, SyMRP was preincubated for 10 min before ATP addition. Following preincubation, ATP was added to a final concentration of 0.1 mM, and the mixture was then incubated for 0 to 10 min. Preincubation and incubation were conducted at 37 °C or 42 °C. Every 2 min, 80 μL aliquots were taken from the reaction mixture, and 20 μL of the Working Reagent of the Malachite Green Phosphate Assay Kit (BioAssay Systems, Hayward, CA, USA) was added. After incubation at room temperature for 32 min, the absorbance at 600 nm was measured via a GloMax DISCOVER (Promega, Madison, WI, USA). The amount of free phosphate was quantified by fitting the obtained absorbance to a standard curve, and the ATPase activity (μM Pi/min/mg protein) was calculated. Measurements were also performed on samples with ATP alone, without the addition of SyMRP, to quantify the amount of phosphate generated by the natural degradation of ATP and subtracted to correct for it.

### 4.6. Protein Aggregation Measurements

Thermal aggregation of citrate synthase from the porcine heart (CS) (Sigma Aldrich, St. Louis, MO, USA) was monitored by measuring light scattering at 500 nm with a spectrofluorometer (FP-6500, JASCO, Tokyo, Japan) at 43 °C as described previously. Native CS (0.1 µM) was incubated in TKM buffer with or without SyMRP (0.6 and 1.2 µM). The effect of nucleotides was examined by the addition of 1 mM ATP or ADP. BSA was used as a control protein. The assay buffer was preincubated at 43 °C and continuously stirred throughout the measurement.

### 4.7. Size Exclusion Chromatography on HPLC

Size exclusion chromatography on HPLC (SEC) was performed with a gel filtration column (SB-804HQ, Showa Denko, Tokyo, Japan) using an HPLC system, PU-1580i, connected to a UV-4075 UV–visible absorbance detector (JASCO) using the buffer (50 mM Tris-HCl pH 7.5, 0.1 mM EDTA, 150 mM NaCl) at a flow rate of 0.5 mL/min. The proteins were monitored by the absorbance at 215 nm. The molecular mass was calculated using the following molecular markers: thyroglobulin (bovine), γ-globulin (bovine), ovalbumin (chicken), myoglobin (horse), and vitamin B12.

### 4.8. Analytical Ultracentrifugation (AUC) Measurements

AUC measurements were carried out with a ProteomeLab XL-I (Beckman Coulter, Indianapolis, IN, USA). SyMRP solutions were prepared at 2.0 mg/mL in 50 mM Tris-HCl buffer (pH 8.0) containing 200 mM NaCl with and without 1 mM ATP. The samples were loaded into a cell with a 12 mm optical-pathlength. All of the measurements were carried out using Rayleigh interference optics at a rotor speed of 60,000 rpm and a temperature of 25 °C. The time evolution of the sedimentation data was analyzed with the multicomponent Lamm formula using SEDFIT software (version 15.01c) [21]. The weight concentration distribution *c*($s_{20,w}$) was subsequently obtained as a function of the sedimentation coefficient, which was normalized to the value at 20 °C in pure water $s_{20,w}$. The molecular mass *M* of each component was calculated using the following equation:(1)$$M={\left[\frac{6\pi \eta N_{A}}{\left(1-\rho \bar{v}\right)}\right]}^{1.5}{\left(\frac{3\bar{v}}{4\pi N_{A}}\right)}^{0.5}{\left(\frac{f}{f_{0}}\right)}^{1.5}{s_{20,w}}^{1.5},$$
where ρ, η, $\bar{v}$,$\;N_{A}$, and *f*/*f*_0_ are the density of pure water at 20 °C, viscosity of pure water at 20 °C, partial specific volume, Avogadro number, and frictional ratio, respectively.

### 4.9. Small-Angle X-ray Scattering (SAXS) Measurements

SAXS measurements were carried out with a laboratory-based instrument NANOPIX (Rigaku, Tokyo, Japan) for SyMRP solutions at 2.0 mg/mL in 50 mM Tris-HCl buffer (pH 8.0) containing 200 mM NaCl with and without 1 mM ATP. Prior to SAXS measurement, the solutions were firstly centrifugated at 17,000× *g* and 4 °C for 10 min. Subsequently, the solution was filtered by spin filtering with a pore size of 0.22 μm. Point-focused X-rays at a wavelength of 1.54 Å were generated by a Cu Kα source MicroMAX-007 HFMR (Rigaku). Scattered X-rays were counted with a HyPix-6000 hybrid photon counting detector (Rigaku). The sample-to-detector distances were set to 1330 mm and 350 mm, covering a *q* range of 0.01 Å^−1^ ≤ *q* ≤ 0.50 Å^−1^, where *q* is the magnitude of the scattering vector. Two-dimensional scattering patterns were converted to one-dimensional scattering profiles using SAngler 2.1.69 software [22]. All of the measurements were carried out at 25 °C.

### 4.10. Molecular Modeling

Molecular modeling was performed with AlphaFold3 [23]. The SAXS profile for the molecular model was calculated with CRYSOL 3.0 in the ATSAS software package [24].

### 4.11. AFM Observation

AFM observation was performed using a self-constructed high-speed AFM apparatus [25,26]. The samples were applied to Ni^2+^-coated mica for 5 min, followed by washing with 50 mM Tris-HCl, pH 7.5. The scan area was 30 × 30 nm^2^ or 40 × 40 nm^2^ with 60 × 60 pixels, and the scan rate was 0.5 s/frame. The cantilevers were used for high-speed scanning. A low-pass filter was used on the images to remove noise.

### 4.12. Electron Microscopy

The samples were applied to carbon-coated copper grids and negatively stained with 2.5% (*w*/*v*) uranyl acetate. Micrographs were recorded at a magnification of 50,000× with a JEM-1400 transmission electron microscope (JEOL, Tokyo, Japan) operated at 100 kV.

## Figures and Tables

**Figure 1 ijms-25-09955-f001:**
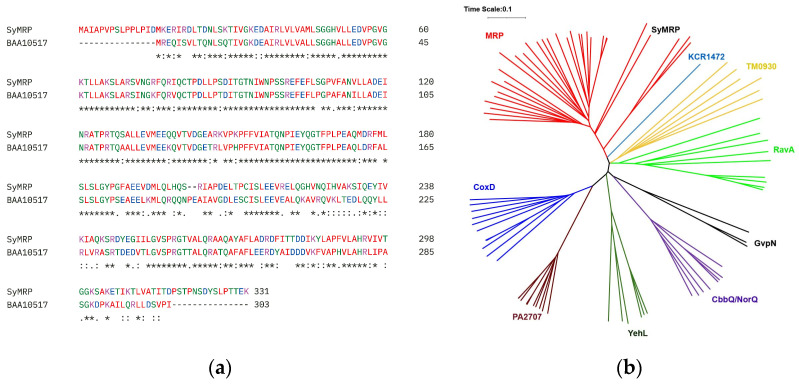
Amino acid sequence alignment (**a**) and phylogenetic tree of SyMRP (**b**). Amino acid sequence alignments were performed via ClustalOmega. A phylogenetic tree was generated by iTOL.

**Figure 2 ijms-25-09955-f002:**
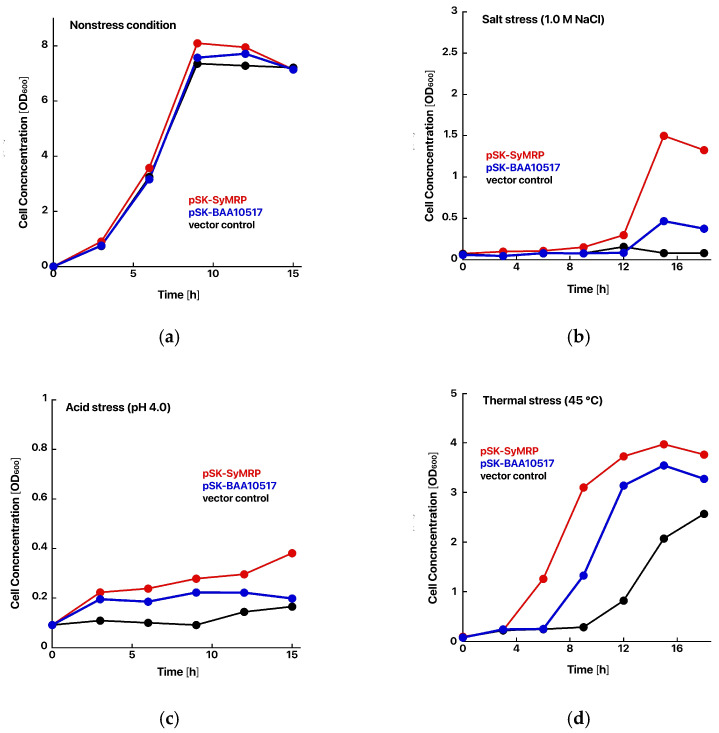
Effects of SyMRP on the growth of *E. coli* under various stress conditions. (**a**–**d**) Growth curves under various stress conditions. (**a**) Nonstress condition. (**b**) Salt stress condition (1.0 M NaCl). (**c**) Acid stress condition (pH 4.0). (**d**) Thermal stress condition (45 °C). Red circle: pSK-SyMRP; blue circle: pSK-BAA10517; black: vector control. (**e**–**g**) Spot tests. 1: pSK-SyMRP, 2: pSK-BAA10517, 3: vector control. (**e**) Nonstress condition. (**f**) Salt stress condition (0.8 M NaCl). (**g**) Thermal stress condition (50 °C, 30 min).

**Figure 3 ijms-25-09955-f003:**
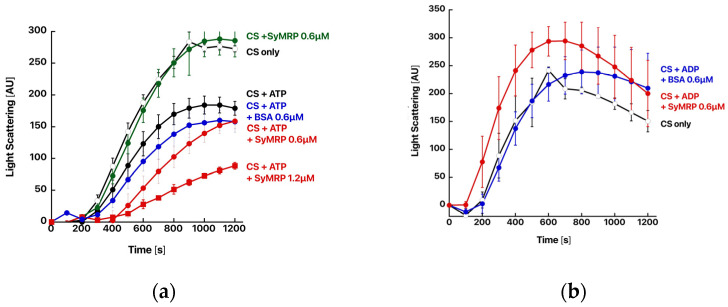
Chaperone function of SyMRP. The thermal aggregation of CS from the porcine heart was monitored by measuring light scattering at 500 nm with a spectrofluorometer at 43 °C. CS (0.1 µM, monomer) was incubated in assay buffer with or without SyMRP and BSA at the specified concentration. To examine their effects, 1 mM ATP (**a**) or ADP (**b**) was added.

**Figure 4 ijms-25-09955-f004:**
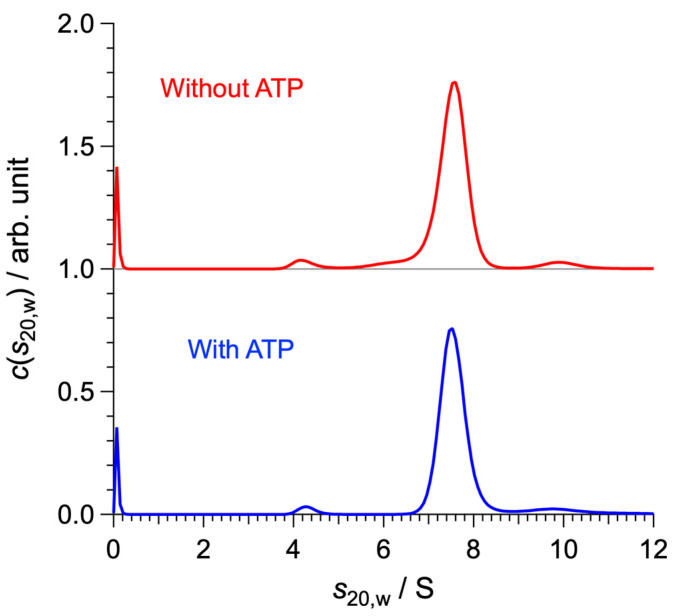
AUC results of SyMRP solutions at 25 °C. The red and blue lines show the concentration distributions of molecules as a function of the sedimentation coefficient for the solutions without and with ATP, respectively.

**Figure 5 ijms-25-09955-f005:**
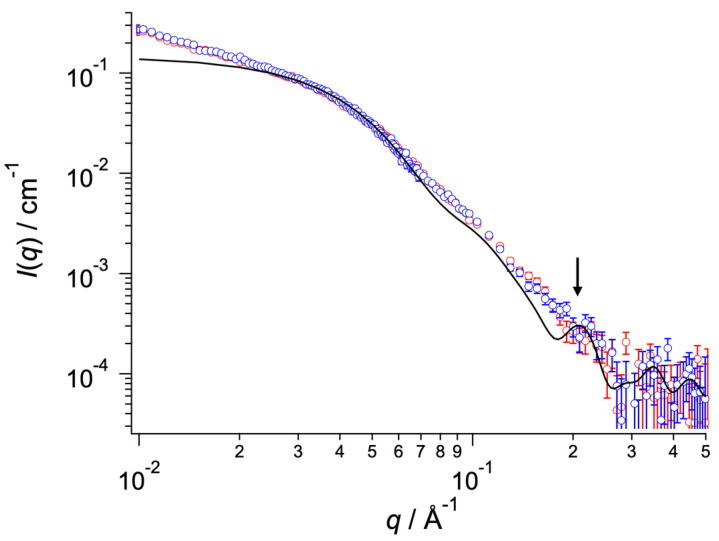
SAXS profiles of SyMRP solutions at 25 °C. The red and blue circles show the SAXS profiles of the solutions without and with ATP, respectively. The solid black line indicates the calculated scattering profile for the hexameric model predicted by AlphaFold 3. The arrow represents the peak for the calculated scattering profile at *q* ~ 0.2 Å^−1^.

**Figure 6 ijms-25-09955-f006:**
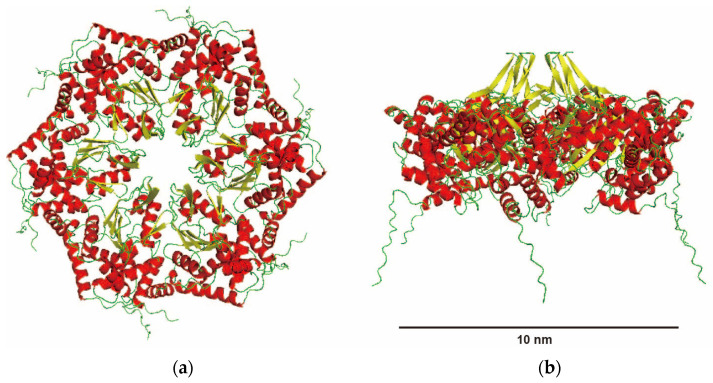
Hexameric structure of SyMRP predicted by AlphaFold 3. (**a**) Top view, (**b**) side view.

**Figure 7 ijms-25-09955-f007:**
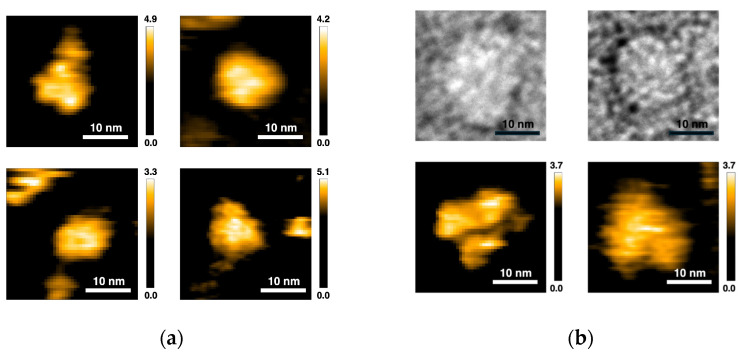
AFM and TM images of SyMRP. (**a**) Images of small oligomers observed by AFM. (**b**) Image of probable hexamers observed by TEM (**top**) and AFM (**bottom**).

**Table 1 ijms-25-09955-t001:** Sedimentation coefficient *s*_20,w_, molecular mass *M*, and weight fraction *w* of SyMRP oligomers observed in AUC.

	*s*_20,w_/S	*M*/kDa	*w*/%
Without ATP	4.18	90.4	4.0
	7.60	221	91.4
	9.87	327	4.6
With ATP	4.26	93.2	2.6
	7.53	218	90.9
	9.72	320	6.5

## Data Availability

The original contributions presented in the study are included in the article/Appendix A; further inquiries can be directed to the corresponding author.

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
