# Peer review of "Molecular Characterization of the MoxR AAA+ ATPase of Synechococcus sp. Strain NKBG15041c"

_ijms, 2024, doi:10.3390/ijms25189955_

Round 1

Reviewer 1 Report

Comments and Suggestions for Authors

Mano et al. isolated a stress tolerance-related gene from Synechococcus sp. and confirmed this gene expresses a MoxR AAA+ ATPase protein, SyMRP. The authors revealed that SyMRP showed weak ATPase activity and also chaperone activity from the stress resistance experiment in E. coli. By further examined by various methods – analytical ultracentrifugation (AUC), small-angle X-ray scattering (SAXS), atomic force microscopy (AFM), and transmission electron microscopy (TEM), the authors revealed that this protein does not have a canonical ring structure of MoxR AAA+ ATPase, despite the predicted structure by AlphaFold 3 displayed the canonical flat ring structure.

Although the experiments for functional characterization were nicely done, it is difficult to find the connection between their structural analysis and the non-canonical conformation they claimed. There are several points of concern that need to be addressed.

1.      The authors compared the sequence of SyMRP with another protein, BAA10517, and probably can do with some other known AAA+ ATPases as well. Based on that alignment, can the authors design mutational assays that abolish the ATPase activity of SyMRP and support the chaperon function in the stress tolerance experiment?

2.      The authors set the analytical ultracentrifugation (AUC) condition with or without ATP and concluded that there are almost no changes in their conformation. My concern in this experiment is that the ATP could be potentially hydrolyzed while running the experiment because of its ATPase activity. Can the authors test this with ATP analogs that are resistant to hydrolysis by ATPases?

3.      In Figure 5, the low-concentration SAXS profile is showing some aggregation of the protein. How did the authors prepare and treat their samples? Was the sample filtered before SAXS analysis? It is also surprising that the experiment was done only with a single concentration.

4.      As the authors obtained and analyzed the SAXS signals, I am assuming that they could obtain the Dmax values as well as the low-resolution 3D envelope of SyMRP. How are the values and shapes different from the AlphaFold 3 predicted structure and/or AFM and TEM analysis results?

5.      Regarding the SAXS experiment, the method section is insufficiently described the detail and there is missing information about the concentration of the proteins used for the experiment and how the sample was treated. Please the authors describe more thoroughly.

6.      In Figure 6, can the authors measure the dimensions of the AlphaFold predicted structure, compare them with the AFM and TEM analysis results, and describe about the difference in the text? This is crucial because the authors claimed that the SyMRP does not have a canonical ring structure, but they do not have a proper comparison supporting this idea in the text.

Comments on the Quality of English Language

7.      Line 148, ‘unimer’ should mean ‘monomer’

Author Response

To Reviewer 1

Thank you very much for your kind suggestions regarding our manuscript. We have revised it according to your comments. Details of the revisions and responses to the queries are as follows:

Q1. The authors compared the sequence of SyMRP with another protein, BAA10517, and probably can do with some other known AAA+ ATPases as well. Based on that alignment, can the authors design mutational assays that abolish the ATPase activity of SyMRP and support the chaperon function in the stress tolerance experiment?

A1. Thank you very much for your valuable suggestion. Yes, it is important to understand the effect of ATPase activity of SyMRP on stress tolerance. To investigate this, we need to construct a recombinant Synechococcus sp. NKBG15041c. We plan to undertake this in our next project.

Q2. The authors set the analytical ultracentrifugation (AUC) condition with or without ATP and concluded that there are almost no changes in their conformation. My concern in this experiment is that the ATP could be potentially hydrolyzed while running the experiment because of its ATPase activity. Can the authors test this with ATP analogs that are resistant to hydrolysis by ATPases?

A2. Although we observed ATPase activity of SyMRP, it is relatively weak as observed in other MoxR AAA+ ATPase. ATPase activity of SYMRP was about 10 nmol Pi/min/ 1mg SyMRP at 37°C. 1mg SyMRP is 27.8 nmol monomer. Thus, turnover rate is only 0.36 min-1. Since the ATP concentration in the experiments was 1mM which is significantly high than SyMRP. So, we don’t think that ATP is consumed during experiments.

Q3. In Figure 5, the low-concentration SAXS profile is showing some aggregation of the protein. How did the authors prepare and treat their samples? Was the sample filtered before SAXS analysis? It is also surprising that the experiment was done only with a single concentration.

A3. We used the same protein sample as that used for the AUC experiment. The details of the sample preparation and treatment (centrifugation and filtration) prior to SAXS measurement are as follows: “Prior to SAXS measurement, the solutions were firstly centrifugated at 17,000 g and 4 °C for 10 minutes. Subsequently, the solution was filtered by spin-filter with the pore size of 0.22 μm.” This description has been added to the Materials and Methods section.

We have not explained reason for adopting the 2.0 mg/mL for SAXS measurement in our previous manuscript, hence we would like to explain the reason in this respond. In principle, the observable scattering intensity (I(q)) by SAXS is given by a following equation.

I(q) = |F(q)|2 S(q),                                                           eq. (1)

where F(q) and S(q) correspond to the concentration-independent form factor and the inter-particle interference structure factor, respectively. For the reliable determination of solution structure of SyMRP, we should select an appropriate protein concentration which is free from the contribution of S(q). Based on Percus-Yevick model, we calculated S(q) at 4 different protein concentrations (refer to Fig. R1). The contribution of S(q) at the lowest q region is higher than 0.95 at the protein concentration lower than 5.0 mg/mL, meaning almost negligible contribution of S(q). Considering the balance between counting statistic and less contribution of S(q), we determined to adopt 2.0 mg/mL for the present work.       

Figure R1. Inter-particle interference factor S(q) calculated with Percus-Yevick model at four different concentrations.

Q4. As the authors obtained and analyzed the SAXS signals, I am assuming that they could obtain the Dmax values as well as the low-resolution 3D envelope of SyMRP. How are the values and shapes different from the AlphaFold 3 predicted structure and/or AFM and TEM analysis results?

A4. In this study, we mainly focused to discuss whether the predicted structure by AF3 was observable in solution structure. As we have already reported in our previous version of manuscript, the predicted structure by AF3 was not observed by SAXS profiles. It should be stressed that our discussion is limited to the higher-q range where the contribution of the large aggregates was relatively small. Following the suggestion by the reviewer, we also calculated pair-distance function P(D) and the low-resolution 3D envelope. Since the upturn due to aggregates were observed for SAXS profiles, we used two procedures for the evaluation of Dmax and resulting low-resolution 3D envelopes. As a first approach, we used the whole q region (q ≥ 0.01 Å-1). As a second approach, we only used the q region higher than 0.03 Å-1 where the contamination by aggregates is smaller. The results from two approaches are summarized in Fig. R2. It is evident that both Dmax values and resulting low-resolution 3D envelopes are severely affected by the selection of q region. We therefore determined not to discuss the detail of Dmax values and resulting low-resolution 3D envelopes in this article. To obtain the reliable scattering profile without the effect of aggregates, we are planning to perform size-exclusion chromatography-SAXS (SEC-SAXS) measurement for the present samples. At least 3.0 mg of SyMRP are required for SEC-SAXS measurement, hence we have started to prepare enough amount of sample. The detail of the 3D structure with the reliable scattering profile will be reported in our future work.

Figure R2. Pair-distance functions P(D) calculated using the scattering profile in (a) q ≥ 0.01 Å-1 and (b) q ≥ 0.03 Å-1. Red and blue lines show P(D) for SyMRP without and with ATP, respectively. Low-resolution 3D model calculated using the scattering profile in (c) q ≥ 0.01 Å-1 and (d) q ≥ 0.03 Å-1. Red and blue envelopes show the model for SyMRP without and with ATP, respectively. Green ribbon represents the structure of SyMRP predicted by AF3.

Q5. Regarding the SAXS experiment, the method section is insufficiently described the detail and there is missing information about the concentration of the proteins used for the experiment and how the sample was treated. Please the authors describe more thoroughly.

A5. As responded in A3, we added the detail of the sample preparation and treatment prior to SAXS measurement in Materials and Methods section.

Q6. In Figure 6, can the authors measure the dimensions of the AlphaFold predicted structure, compare them with the AFM and TEM analysis results, and describe about the difference in the text? This is crucial because the authors claimed that the SyMRP does not have a canonical ring structure, but they do not have a proper comparison supporting this idea in the text.

A6. Thank you very much for your important comments. We have also confused with the results. Since SAXS data has clearly deny a symmetrical ring structure, we think SyMRP hexamer does not take the symmetric ring structure.

The following description was included in Discussion.

“Since all experimental data don’t support AlphaFold 3 prediction, we think that SyMRP does not take the canonical symmetric ring structure.”

Q7. Line 148, ‘unimer’ should mean ‘monomer’

A7. We are sorry for our careless mistake. It was revised.

Reviewer 2 Report

Comments and Suggestions for Authors

Comments to authors

In this review entitled “Structure and function of the MoxR AAA+ ATPase of Synechococcus sp. Strain NKBG15041c”, Mano et al. identified a stress tolerance-related gene from Synechococcus sp. 16 NKBG15041c, which was designated as SyMRP. A series of characterizations were done to investigate the recombinant SyMRP protein. Generally, the methods applied in this study are proper. However, the author must address the following concerns in the manuscript before publishing this paper.

1. On page 4, lines 122-128, the authors found that recombinant SyMRP exhibits ATPase activity. This is a very important conclusion; thus, the standard curve of enzyme activity, as well as pH-plated enzymic activity and temperature-related enzymic activity, should be shown in the manuscript.

2. The author believes that recombinant SyMRP has an ATP-dependent Chaperone function, supported mainly by results in Figure 3. However, the detailed experimental design in Figure 3 has a problem. As shown in the figure legend, only 0.1 um CS was used in the samples; however, 0.6 um or 1.2 um SyMRP was used as 6- or 12-times CS. Thus, how could we say the measured light scattering results (thermal aggregation) are contributed by CS rather than SyMRP?

3. In Figure 3's right panel, ADP-related curves have a strange tendency. I am not sure why the light scattering eventually decreases. However, at least two CS-only curves in the left panel and right panel should be identical.

4. SEC and AUC show that recombinant SyMRP exists as a hexamer. What is the diameter of the hexametric model predicted from AF3? Is this diameter similar to that shown in negative stain TEM? It is hard to imagine how a hexamer was assembled without an axis of C6 symmetry. However, I didn’t see any evidence of the C6 symmetry in the TEM image. Thus, the statement in line 192, “The TEM image revealed the formation of a hexamer, which does not have a canonical ring structure (Fig. 7b).” is improper. At least there is no evidence supporting hexamer from TEM images.

Other small issues:

Now, the readers have to read the figure legend to understand each panel's meaning. That could be simpler and easier.

For example,

In Figure 2, the corresponding stress conditions in panels a-d could be labeled in the figure, not only in the figure legend. “Nonstress condition” in (a). “Salt stress (1.0 M NaCl) in (b).  Acid stress (pH 4.0) in (c).

In Figure 2, e—f, based on my understanding, each row should respond to different conditions; these should be directly marked and labeled in the figure.

Figure 6, a size indicator could be added to show the diameter of the hexamer predicted by AF3.

Comments on the Quality of English Language

Acceptable.

Author Response

Thank you very much for your kind suggestions regarding our manuscript. We have revised it according to your comments. Details of the revisions and responses to the queries are as follows:

Q1. On page 4, lines 122-128, the authors found that recombinant SyMRP exhibits ATPase activity. This is a very important conclusion; thus, the standard curve of enzyme activity, as well as pH-plated enzymic activity and temperature-related enzymic activity, should be shown in the manuscript.

A1. Thank you very much for your critical comment. The results of ATPase assay experiments are shown in Fig. R3. ATPase activity at 37°C and 42°C were calculated from the amount of Pi released from ATP. As the ATPase was not high compared with the spontaneous degradation, the activity was calculated considering spontaneous degradation. Thus, it was difficult to determine detailed kinetic parameters. The calculated ATPase activities at 37°C and 42°C are shown in the test. Although it is suggested to examine ATPase activities at various pH and temperatures, we think the present data are enough. The effects of ATP on CS aggregation assay at the almost same temperature and pH.

Figure R3. ATPase activity profiles at 37 °C and 42°C

Q2. The author believes that recombinant SyMRP has an ATP-dependent Chaperone function, supported mainly by results in Figure 3. However, the detailed experimental design in Figure 3 has a problem. As shown in the figure legend, only 0.1 µm CS was used in the samples; however, 0.6 µm or 1.2 µm SyMRP was used as 6- or 12-times CS. Thus, how could we say the measured light scattering results (thermal aggregation) are contributed by CS rather than SyMRP?

A2. Thank you very much for your critical comment. The molar concentration is at the subunit level. Since SyMRP exists as a hexamer, 6-fold or 12-fold of CS means that one or two SyMRP hexamers function as one CS monomer. In general aggregation assay of chaperones, the ration is not excessive. At the beginning of the aggregation experiments, we confirmed that SyMRP itself does not aggregate upon heat treatment. Additionally, 0.6 µM SyMRP had no effect on the aggregation of CS, as shown in Fig. 3a.

Q3. In Figure 3's right panel, ADP-related curves have a strange tendency. I am not sure why the light scattering eventually decreases. However, at least two CS-only curves in the left panel and right panel should be identical.

A3. Thank you very much for your critical comment. In the CS aggregation assay, a decrease in light scattering occurs often. Initially, heat-denatured proteins form small aggregates that scatter visible light. As aggregation progresses, these aggregates combine to form larger aggregates, which leads to a decrease in light scattering. For example, precipitated protein aggregates do not scatter light.

It is difficult to control protein aggregation. CS was obtained as ammonium sulfate precipitants and used after dialysis. Although we try to obtain uniform CS, it contains denatured or aggregated ones. The aggregation speed changes as the abundance of denatured or aggregated CS. The amounts of denatured or aggregated ones was trace and difficult to control. So, the aggregation profiles change a lot.

Q4. SEC and AUC show that recombinant SyMRP exists as a hexamer. What is the diameter of the hexametric model predicted from AF3? Is this diameter similar to that shown in negative stain TEM? It is hard to imagine how a hexamer was assembled without an axis of C6 symmetry. However, I didn’t see any evidence of the C6 symmetry in the TEM image. Thus, the statement in line 192, “The TEM image revealed the formation of a hexamer, which does not have a canonical ring structure (Fig. 7b).” is improper. At least there is no evidence supporting hexamer from TEM images.

A4. Thank you very much for your important comment. I am sorry for my ambiguous description. The description was changed to “The TEM image seems to be a flat ring structure. But, its resolution was poor to observe C6 symmetry”

Q5: Now, the readers have to read the figure legend to understand each panel's meaning. That could be simpler and easier.

For example,

In Figure 2, the corresponding stress conditions in panels a-d could be labeled in the figure, not only in the figure legend. “Nonstress condition” in (a). “Salt stress (1.0 M NaCl) in (b).  Acid stress (pH 4.0) in (c).

In Figure 2, e—f, based on my understanding, each row should respond to different conditions; these should be directly marked and labeled in the figure.

Figure 6, a size indicator could be added to show the diameter of the hexamer predicted by AF3.

A5: Thank you very much for your kind suggestion. We have revised them as you indicated.

Round 2

Reviewer 1 Report

Comments and Suggestions for Authors

The authors adequately addressed my concerns and improved figures and methods. The only thing that I would like to suggest is changing the title. As the current works and conclusions do not provide insights into the structure of this protein, 'structure and function' could be misread by the readers. 

Author Response

Thank you very much for your important comment. We have changed the title to "Molecular characterization of the MoxR AAA+ ATPase of Syn-echococcus sp. strain NKBG15041c."

Reviewer 2 Report

Comments and Suggestions for Authors

Thanks for your revision. The manuscript is much better now.

Author Response

Thank you very much for your kind consideration. We could improve our manuscript by your important comments.